# Mechanism and Application of Layered Grouting Reinforcement for Fractured Coal and Rock Roadway

Ze Liao * and Tao Feng

School of Resource & Environment and Safety Engineering, Hunan University of Science and Technology, Xiangtan 411201, China
* Correspondence: leo_hnust@163.com

**Abstract:** This paper takes the ZF3806 working face of Shuiliandong Coal Mine in Binxian County, Shaanxi Province as the engineering background. Aiming at the problems of the development of surrounding rock cracks and roof breakage encountered in the process of roadway excavation and support and based on the composite beam theory, the method of layered grouting reinforcement of roadways is proposed according to the deformation and failure of the roadway roof and the internal drilling conditions. At the same time, combined with the splitting grouting mechanism, the roadway is strengthened and supported by layered grouting of "shallow bolt grouting + deep cable grouting". The "shallow" and "deep" form a complete and stable composite beam support structure. After grouting, the bending moments of "shallow" and "deep" support beams decrease by $20.78 \times 10^6$ N·m and $26.50 \times 10^6$ N·m, respectively. The support scheme is applied to the field test, and the grouting effect is analyzed and monitored. The research results show the layered grouting support scheme of "shallow bolt grouting + deep cable grouting" can significantly improve the structural integrity of the roadway roof. The displacement of the two sides is within the controllable range, and the support role of the bolt and cable is entirely played through grouting. The roof displacement of the roadway is reduced by 65% on average, and the bolt failure and steel belt fracture are significantly reduced, which effectively controls the deformation and damage of the roadway and reduces the maintenance cost of the roadway while ensuring safe mining. The study's findings could be useful in treating broken surrounding rock in other coal mine roadways.

**Keywords:** fractured coal; roadway support; composite beam; layered grouting; stress gradient



## 1. Introduction

Coal is a major energy source and plays an important role in economic development [1–4]. As a non-renewable energy source [5–7], the safe and efficient mining of coal is of great significance. In the process of roadway support in the coal mine and due to the development of primary fissures in the surrounding rock of the roadway [8–10], the surrounding rock is broken after roadway excavation, which leads to difficulties in roadway support and makes it easy to cause roadway deformation and damage after support. For this type of roadway, the available bolt and cable support cannot achieve the expected support effect and cannot effectively utilize the mechanical properties of the bolt and anchor cable. It has no constraint on the roadway, resulting in a poor support effect. Therefore, a bolt and cable combined with grouting reinforcement support have become an effective way to control the surrounding rock of the roadway.

After years of roadway support research, domestic and foreign scholars have accumulated much theoretical and practical experience in the theory and technology of bolt-grouting reinforcement. At present, the depth of coal mining is increasing [11–13]. Kang Hongpu et al. [14,15] improved the surrounding rock body's mechanical properties while ensuring the anchor bolt's and cable's anchoring effects through grouting modification technology for the 1000-m-deep mine roadway and obtained a good support effect.

Liu Quansheng et al. [16,17] used the anchor-grouting method to control the deformation of the surrounding rock of the roadway and studied the influence of grouting on the mechanical properties of fractures. Xie Shengrong et al. [18,19] put forward a comprehensive control technology of anchor grouting with new cement-grouting materials and high-elongation anchor grouting by analyzing the stability of surrounding rock before and after the tunnel grouting, which successfully controlled the deformation and failure of the tunnel. Yu Weijian et al. [20–22] put forward bolt-, anchor cable-, and two-step-grouting shell support technology to realize internal and external bearings for a roadway with large deformation fissure rock mass and achieved a good support effect. At the same time, some researchers have conducted much research on the interaction between bolts and surrounding rock [23–25]. Wang Qi et al. [26,27] used grouting anchor cable active support instead of passive hydraulic prop support to solve the problem of large deformation and complex conditions of a deep mine roadway, and the roof was effectively controlled. Salimian et al. [28–30] carried out a systematic study on the mechanical behavior of grouted fissured rock mass and the characteristics of grouting fissures. At the same time, some experts also carried out much theoretical and experimental research on the flow and diffusion of grouting fluid in the cracks in the grouting reinforcement support [31–34]. When the surrounding rock of the roadway is relatively broken, the bolt and anchor cable combined with grouting support is an effective support method. Different support schemes and structures can also adapt to different geological conditions.

Support problems are encountered in the mining roadway of the ZF3806 working face in Shuiliandong Coal Mine in Binxian County, Shaanxi Province. This paper proposes the roof of the mining roadway should be supported by layered grouting with bolts and cables to improve the integrity of the roadway's surrounding rock so the roadway roof can form a stable composite beam support structure and ensure the safe and efficient mining of the mining face. To a certain extent, field tests are expected to provide theoretical and practical guidance for roadway support of the same type of fractured surrounding rock.

## 2. Engineering Overview

### 2.1. General Situation of the Coal Roadway

The ZF3806 working face of Shuiliandong Coal Mine in Binxian County, Shaanxi Province, is located at the +770 level in the south of the third mining area of the coal mine. The distance between the transportation roadway of the working face and the goaf of the ZF3804 working face is 24 m. The south side of the air roadway of the working face is the boundary of the Shuiliandong coal mine. The design strike length of the roadway is 905 m, the buried depth of the roadway is 320 m~370 m, and the average buried depth is 335 m. The average value of Coal Seam 4 # is 6.4 m, and the vertical joints and fissures of the coal seam are developed. Figure 1 depicts the occurrence conditions of the coal seam's roof and floor.

### 2.2. Deformation and Failure Characteristics of Roadway

According to the field investigation and analysis, the roadway under the original support scheme was seriously deformed and damaged. As shown in Figure 2, the deformation and damage mainly occurred in the roof, characterized by the broken roof, falling off and failure of bolts, uneven deformation of the roof, bending and fracture of the steel belt, roof leakage, and severe roof subsidence. Cracks and local bulges were visible on both sides of the roadway, and the displacement on both sides changed significantly. The workload of roadway maintenance in the later stage was significant, and there were certain safety risks. The main reasons were as follows: the surrounding rock of the roadway was weak, fissures were developed, the degree of fragmentation was high, the support strength was insufficient, and the integrity was poor.

| Lithology | Thickness (m) | Lithology description | Columnar lithology |
|---|---|---|---|
| Siltstone | 3.0 | Gray, mainly siltstone, mixed with thin layer and stripe of fine sandstone. | |
| Fine sandstone | 7.6 | Light gray - gray white, with developed oblique bedding and flat fracture. | |
| Siltstone | 6.5 | Gray, dense and brittle, containing plant fossils and horizontal bedding. | |
| Fine sandstone | 1.4 | Light gray, fine grain structure, argillaceous cementation. | |
| Siltstone | 1.2 | Light gray, silty structure, dense, brittle. | |
| Silty fine sandstone | 2.5 | Light gray, horizontal and wavy bedding, containing plant fossils. | |
| Mudstone | 1.0 | Dark gray, delicate, brittle, with smooth fracture surface. | |
| 4# coal | 6.4 | Black, mainly dark coal, with developed joint fissures, filled with calcite veins. | |
| Aluminous mudstone | 2.0 | Light gray, lumpy, relatively soft, uneven fracture. | |

**Figure 1.** Occurrence conditions of the roof and the floor of the coal seam.

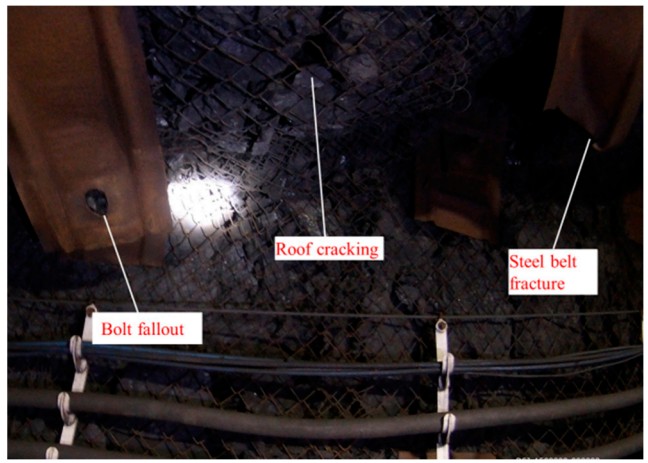
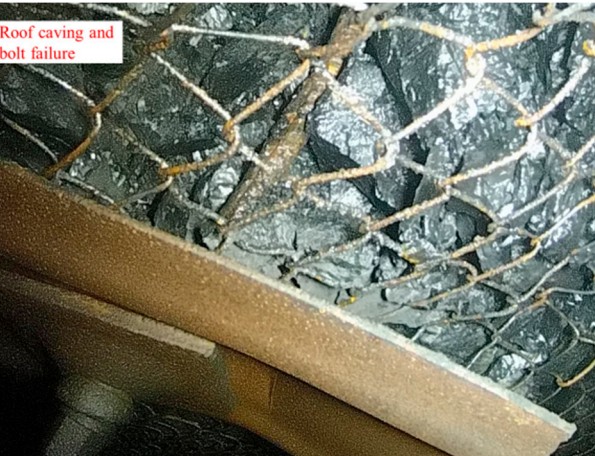

**Figure 2.** Deformation and failure conditions of the roadways.

## 3. Fracture State of Roadway Roof and Its Rock Mechanics Characteristics

### 3.1. Core Drilling

The mining roadway roof of the ZF3806 working face was drilled and cored for 5 m. Due to the fragmentation of the coal body on the roof surface, drilling began 1 m above the roof surface. As shown in Figure 3, the roadway top coal was broken, and the rock core was loose. The mudstone layer 1 m above the coal seam was also relatively broken, while the sandstone layer upward was more and more complete, and there was no obvious fracture in the rock core. Therefore, the crushing range of the roadway was mainly concentrated in the first 3 m of the roof, the sandstone layer behind was relatively complete, and no obvious fracture was found in the rock core.

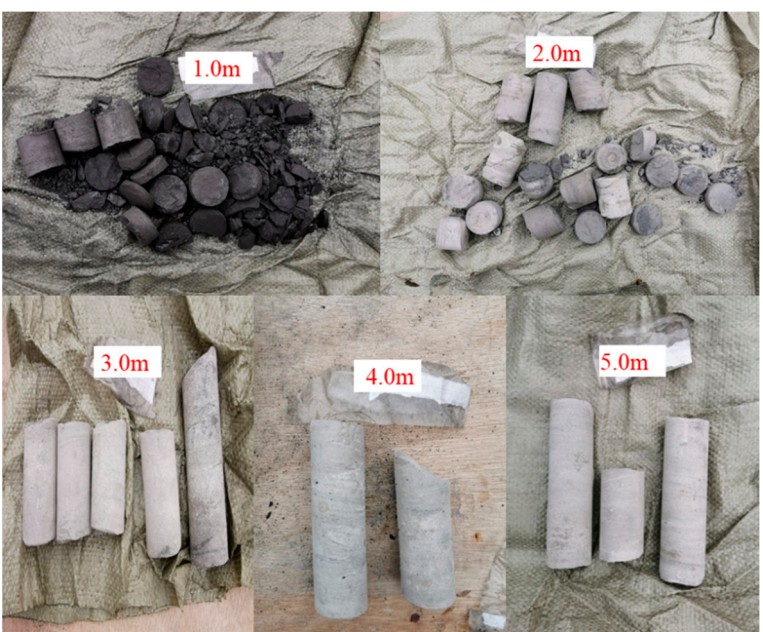

**Figure 3.** Characteristics of rock core.

### 3.2. Rock Mechanics Experiment

According to the research content, the test rock sample was taken from the ZF3806 working face of Shuiliandong Coal Mine in Bin County, Shanxi Province. The rock samples were roof coal, mudstone, silty fine sandstone, and siltstone. Through on-site sampling, the rock samples were transported from the underground to the ground, and coal cores and rock cores were drilled in the laboratory with a coring machine [35–38]. Then, the end was cut and polished with an end grinder. All standard specimens prepared were Φ 50 mm × 100 mm; the standard specimen processing equipment and process are shown in the Figure 4. Rock specimens were processed into standard specimens, and uniaxial compression tests were carried out. Because the mudstone layer was thin and fractured, no good rock specimens were obtained. It was processed into coal rock combination test pieces with some coal specimens to explore the collective failure of mudstone and coal.

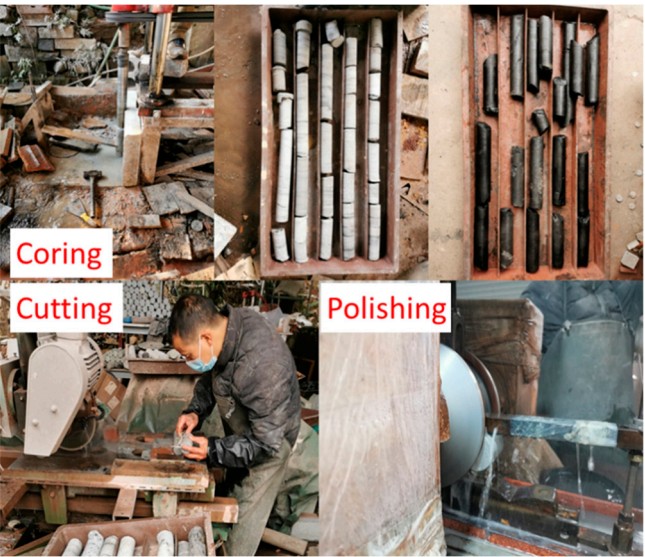

**Figure 4.** Processing of standard specimens.

The stress–strain curve and failure of rock specimens after the uniaxial compression test are shown in Figure 5. The stress–strain curves of all specimens had evident stages of compaction, elasticity, yield, and failure. The failure stage of the coal sample was mainly brittle shear failure. The coal rock assemblage was distinguished by symbiotic double shear failure and numerous fracturing failures. The analysis showed the reason is that there were many microfissures in mudstone and coal. At the same time, the coal sample and coal rock combination exhibited expansion, indicating the roof is subject to obvious crushing and bulking deformation. Sandstone mainly showed shear and double shear failure. The uniaxial compressive strength of the coal sample was 11.42–19.17 Mpa with an average of 14.31 MPa. The uniaxial compressive strength of the coal-rock combination was 7.86–13.49 Mpa with an average of 10.85 MPa. The uniaxial compressive strength of the fine siltstone sample was 18.06–21.13 Mpa with an average of 18.97 MPa. The uniaxial compressive strength of the siltstone sample was 27.92–41.20 Mpa with an average of 35.36 MPa. The strength of the coal rock combination was the lowest and that of siltstone was the highest.

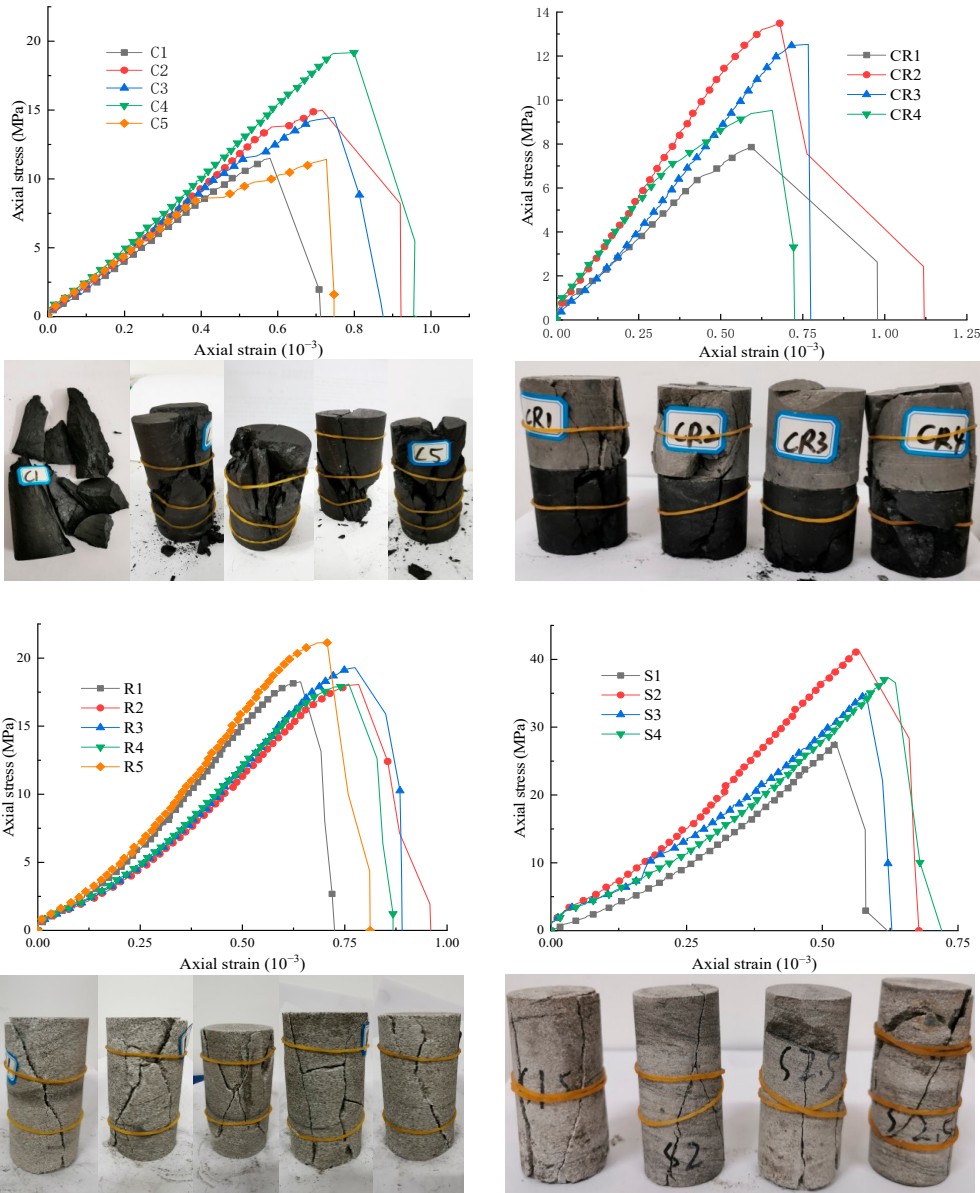

**Figure 5.** Stress–strain curve and failure mode of rock uniaxial compression experiment.

## 4. Mechanism of Layered Grouting Reinforcement for Surrounding Rock of Roadway

### 4.1. Roof Failure Analysis Considering Stress Gradient

According to the roof condition of the mining roadway in ZF3806's working face, after the excavation of the roadway, the initial original rock stress was redistributed, its original stress state changed, and the stress concentration occurred in the surrounding rock of the roadway roof. After the stress of the roof surrounding rock rose, it was destroyed due to its low strength, and the destruction gradually developed from shallow to deep. Figure 6 shows how the shallow and deep fracture zones, plastic zones, elastic zones, and original rock stress zones were formed. This paper defines the roof fracture zone as a "shallow" rock stratum and the plastic zone as a "deep" rock stratum. After the rectangular roadway was excavated, the stress state of its roof changed rapidly from "shallow" to "deep", and there was a stress gradient in the roof, especially in the area where the roof stress is concentrated. The stress gradient in tensor form can be shown as:

$$\eta_{ijk} = \frac{\partial \sigma_{ij}}{\partial x_k} \tag{1}$$

where $\sigma_{ij}$ is the stress tensor, and $x_k$ is the length of the k direction. The stress gradient's absolute value is the stress change speed in this direction—the more significant the stress gradient, the faster the stress change. The area with a significant stress gradient is stress concentration, and the rock mass is more likely to be damaged. When the stress gradient is zero, the stress in this direction does not change.

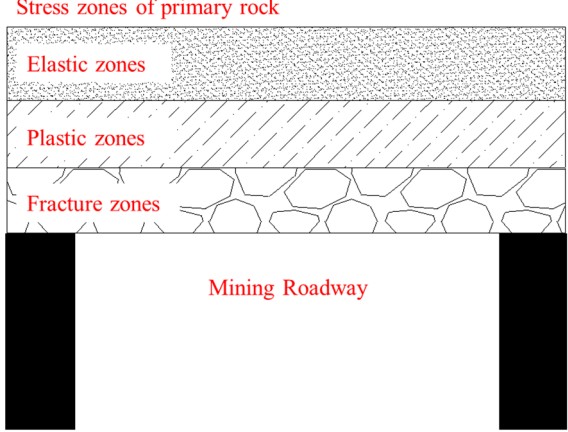

**Figure 6.** Distribution of roadway surrounding rock state.

### 4.2. Mechanism of Roof Layered Grouting Reinforcement

Grouting can improve the fracture structure inside the broken coal and rock mass, increase its internal friction angle and cohesion, and enhance the integrity and strength of the surrounding rock of the roadway. Generally, the split grouting method is used. When the pressure of grout is greater than the shear stress of compaction grouting, the coal bed will split and the grout diffusion radius will increase, forming a more extensive range and a more stable overall structure. The splitting surface produced in the grouting process mainly occurs on the stress surface with the lowest resistance to splitting. As shown in Figure 7, the grout vein network formed by fracturing grouting near the borehole can play a role in extrusion and filling. Squeezing the slurry vein to form the slurry vein to reinforce the surrounding coal and rock mass and improve the stability of the surrounding rock of the roadway reinforces the coal and rock mass.

According to the previous article's coal seam histogram and test analysis, the roof fracture zone mainly occurs in the roof coal seam and mudstone layer. The plastic zone mainly occurs in the silty fine sand stratum. The silt rock stratum has significant strength and is not damaged; thus, it is suitable for anchoring the end rock stratum. Therefore, it is the key to control the stability of the roadway to strengthen the support of the roadway

by using anchor bolts and cables combined with layered grouting to reduce the stress gradient of the roof by layered grouting and improve the self-strength of the rock mass at the same time. The bolt and cable provide active support, and the preload is expanded to increase the vertical stress of the rock stratum. After layered grouting, the stress on the rock stratum is uniform, the stress in the layer is homogeneous, and the stress gradient between layers is reduced so no stress concentration occurs on the roof. At the same time, grouting also increases the cohesion and internal friction angle of the rock mass, reducing its destructive capacity.

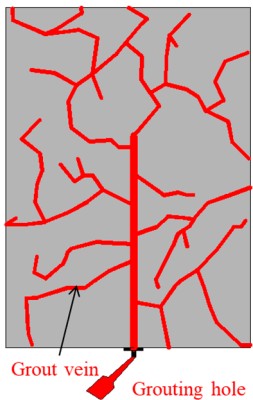

**Figure 7.** Grouting diagram.

As shown in Figure 8, after the roadway excavation, the early setting and early strength grouting fluid is used to support the "shallow" part of the roadway roof so the "shallow" broken coal body can quickly form a rock stratum with a particular strength. At the same time, the internal stress environment is changed by a specific preload, and the stress gradient between rock layers is reduced. Then, the "deep" grouting cable grouting is carried out, and the superfine cement grouting material uses its good fluidity and permeability to make the "deep" fissure-developed coal. Rock masses form a complete rock stratum with long-term stability. After grouting, the roof forms a relatively complete rock stratification, and the "shallow" and "deep" grouting layers are penetrated by cables to form a complete combination. The combination layers squeeze each other, which increases the friction resistance and reduces the rock deflection. The bending strength of the combination increases, and the overall bearing capacity of the surrounding rock mass of the roadway is improved.

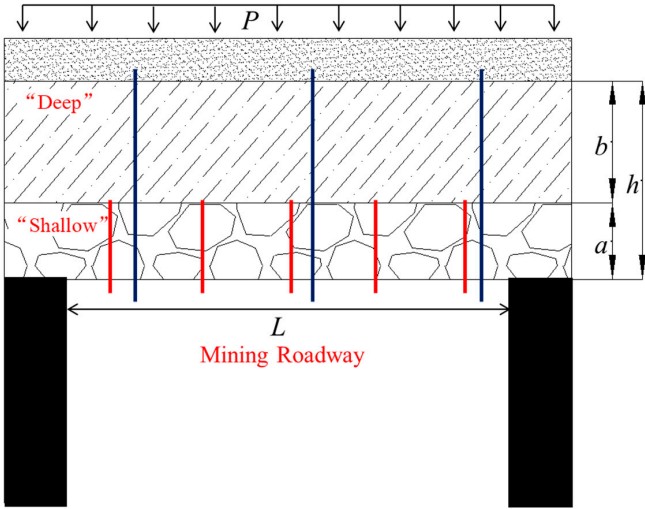

**Figure 8.** Layered grouting support structure.

Figure 9 shows the mechanical model of the roof after layered grouting is established. The roof is simplified as a composite beam, and the grouting area is generally considered elastic. After grouting, the grouting layer will produce support resistance. The support resistance of the "shallow" grouting body is P1 and that of the "deep" is P2. L is the roadway width, a and b are the thicknesses of the "shallow" and "deep" grouting support layers, respectively, and h is the thickness of the composite beam of the grouting layer.

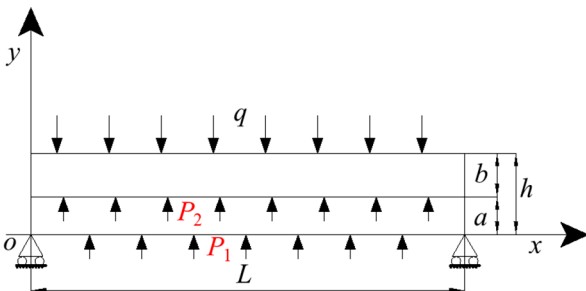

**Figure 9.** Simply supported beam model of roof.

The support resistance of the grouting layer is determined by the strength of the grouting body [39]:

$$P = \frac{[\sigma]}{2}\left[1 - \left(\frac{r}{R}\right)^2\right] \tag{2}$$

where $[\sigma]$ is the uniaxial compressive strength of rock mass after grouting, $r$ is the support radius of the roadway, and $R$ is the support radius of the grouting layer. According to the empirical strength formula of the grouted rock mass [40], the strength of rock mass after grouting can be expressed as:

$$[\sigma] = \frac{3.2543 \times \sigma_c}{\left[2.45 + \log_{10}\sigma_c - \log_{10}q_c\right]^2} \tag{3}$$

where: $\sigma_c$ is the uniaxial compressive strength of rock mass before grouting, and $q_c$ is the strength of grout stone in grouting. Therefore, the bending moment $M_1$ of the "shallow" support body beam can be expressed as:

$$M_1 = \frac{qL^2}{2} - LP_1 \tag{4}$$

The bending moment $M_2$ of the "deep" support body beam can be expressed as:

$$M_2 = \frac{qL^2}{2} - LP_2 \tag{5}$$

The bending moment formula of the beam can be obtained by substituting Equations (2) and (3) into Equations (4) and (5):

$$M_1 = \frac{qL^2}{2} - \frac{1.627\sigma_c L \times \left(2ar + a^2\right)}{\left[2.45 + \log_{10}\sigma_c - \log_{10}q_c\right]^2 \times (r + a)^2} \tag{6}$$

$$M_2 = \frac{qL^2}{2} - \frac{1.627\sigma_c L \times \left(2hr + h^2\right)}{\left[2.45 + \log_{10}\sigma_c - \log_{10}q_c\right]^2 \times (r + h)^2} \tag{7}$$

According to Equations (6) and (7), the main factors affecting the bending moment are the strength of the grouting fluid stone body, the rock mass strength of each rock stratum, and the thickness of the grouting layer. According to the mechanical model analysis, after the roof is strengthened by layered grouting, the bending moments of the "shallow" and "deep" support body beams are reduced, and their abilities to resist bending deformation increase. At the same time, the normal bending stress in the beam of each support body

is also reduced, indicating layered grouting can effectively control the deformation and failure of the roof.

The strength parameters of grouting stones are obtained through laboratory mechanical experiments. The strength of "shallow" grouting material is 30.86 MPa and that of "deep" grouting material is 37.17 MPa. It substitutes the roadway parameters into Equations (6) and (7), and it can be obtained that the bending moment of the "shallow" support body beam decreases by $20.78 \times 10^6$ N·m, and the bending moment of the "deep" support body beam decreases by $26.50 \times 10^6$ N·m.

## 5. Scheme and Application of Grouting Reinforcement Support

### 5.1. Implementation of Grouting Reinforcement Support Scheme

The mining roadway of the ZF3806 working face adopts a rectangular section with a net width of 4.0 m and a net height of 3.5 m. The original support scheme was bolts and cables combined with anchor mesh and a steel belt to strengthen the support, but the effect is poor. Therefore, to ensure the mining face's safe and efficient mining and improve the mining roadway's stability, based on the original support scheme, a reinforcement support method with layered grouting of "shallow bolt + deep cable" as the core is proposed.

1.  Bolt support parameters: As shown in Figure 10, the roadway roof adopts a φ 20 mm × 2200 mm equal strength threaded steel resin bolt and the row spacing of 0.9 m × 1 m. The arch-type high-strength pallet is adopted, and the specification is 150 mm × 150 mm × 10 mm. The extended anchoring method is adopted, and the anchoring length of the bolt is 1050 mm. A W-shaped steel belt support is combined, and a diamond-shaped metal mesh is hung on the whole section with a mesh size of 50 mm × 50 mm. The two sides are arranged with a φ 20 mm × 2200 mm bolt with a row spacing of 1.3 m × 1 m.

2.  Cable support parameters: As shown in Figure 10, the roadway roof adopts a φ 22 mm × 6300 mm hollow grouting cable arranged in a "well type" form with anchor support. The row spacing of the cable is 1.5 m, and the cable is perpendicular to the roof. The distance between the upper side cable of the roof and the upper side is 0.3 m, the distance between the middle cable and the upper side is 1.55 m, and the distance between the lower side cable and the lower side is 0.65 m. The anchoring length of the cable is 1300 mm. The cable is supported by a three-hole steel belt, and the stubble is arranged between steel belts.

3.  Layered grouting reinforcement support: A layered grouting reinforcement support design is implemented for the roadway roof. The "shallow bolt grouting" is based on the original bolt support. The hole depth is 2 m, the grouting holes are located between every two rows of anchor bolts, and the row spacing between grouting holes is 1.8 m × 2 m. When grouting, the borehole shall be used with the grout stop sleeve to make the borehole pass through the fracture zone vertically as far as possible and maximize the diffusion of grouting fluid. The "deep cable grouting" adopts continuous downward grouting carried out in sequence along the roadway. When the design grouting pressure is 10 MPa, the pressure is stabilized for more than 10 min so the grout can spread to the maximum range and the roof rock cracks can be filled as much as possible. The "deep anchor cable grouting" material is superfine cement, and the "shallow anchor bolt grouting" material is GRPC-1 early setting and early strength new material. The "shallow" grouting shall be carried out first and then the "deep" grouting. In the process of "shallow" grouting when there are abnormal phenomena, such as slurry channeling on the surrounding coal wall, the grouting pump shall be stopped in time.

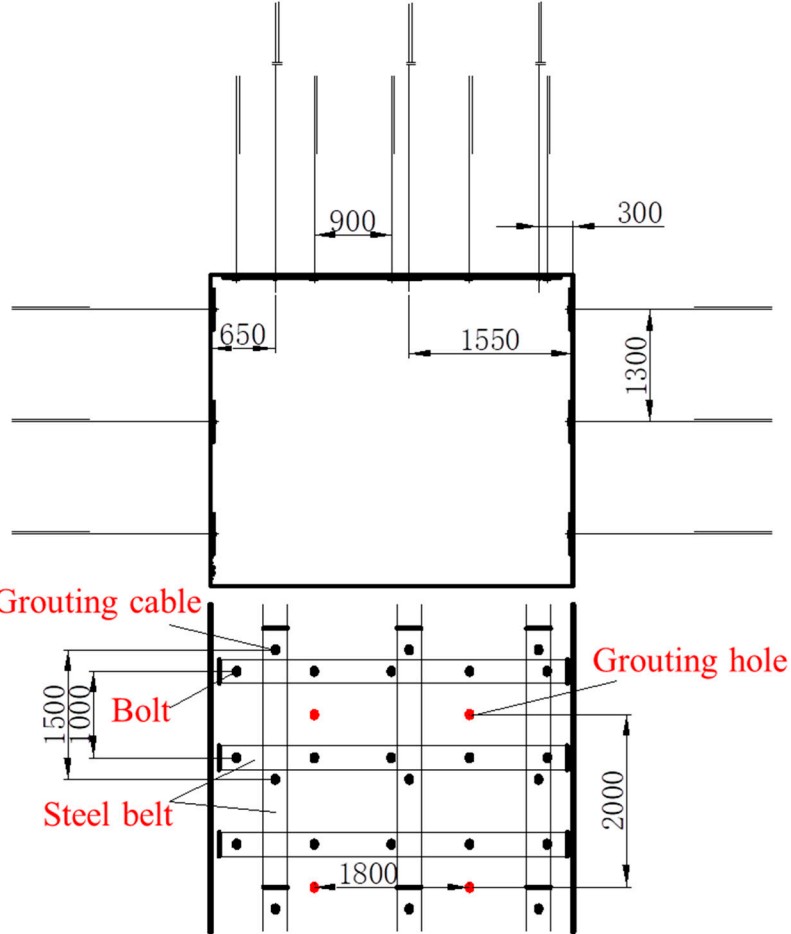

**Figure 10.** Roadway support scheme.

*5.2. Analysis of Roadway Surface Displacement Monitoring Results and Support Evaluation*

(1)　Layout of monitoring points

　　As shown in Figure 11, six displacement monitoring points are designed in the grouting section and the non-grouting section, which are ABCDEF in turn, with a distance of 35 m between points. The section of the grouting reinforcement support scheme is 100 m long, including DEF points, and ABC points are arranged in 100 m of the original support scheme section for a total of 200 m in the test section. The specific arrangement is shown in Figure 11. The "cross" point distribution method is adopted to measure the displacement and deformation of the roadway, and the measurement is conducted every other day.

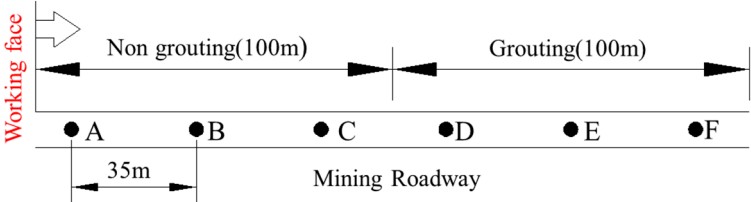

**Figure 11.** Layout of monitoring points.

(2)　Analysis of displacement monitoring results

　　Figure 12 shows the width and displacement change curve of both sides of the mining roadway on the ZF3806 working face. During the monitoring period, the two sides of the ABC point in the non-grouting section moved inward obviously, the displacement changed significantly, and the displacement changed in a curve. At the initial stage, the

distance between the two sides of DEF points in the grouting section has a noticeable displacement change, and the changing trend of the three points is roughly the same, showing a small amount of deformation at first and then tending to be stable. At the same time, sudden inward movement on two sides occurred at six monitoring points. The maximum displacement of the non-grouting section is 0.42 m, and the maximum displacement of the grouting section is 0.2 m. According to the analysis of its causes, in the initial monitoring stage, the mining face is far away from the monitoring point, resulting in an insignificant mining impact; thus, the displacement change of the two sides of the roadway is small. When the mining face continues to advance, the monitoring point enters the mining influence range, causing large deformations on both sides of the non-grouting section roadway. In comparison, the deformation of the two sides of the grouted roadway section is small. The layered grout reinforcement and support can strengthen the stability of the two sides. Only under the influence of periodic pressure, the two sides of the grouting section and the non-grouting section have noticeable displacement changes.

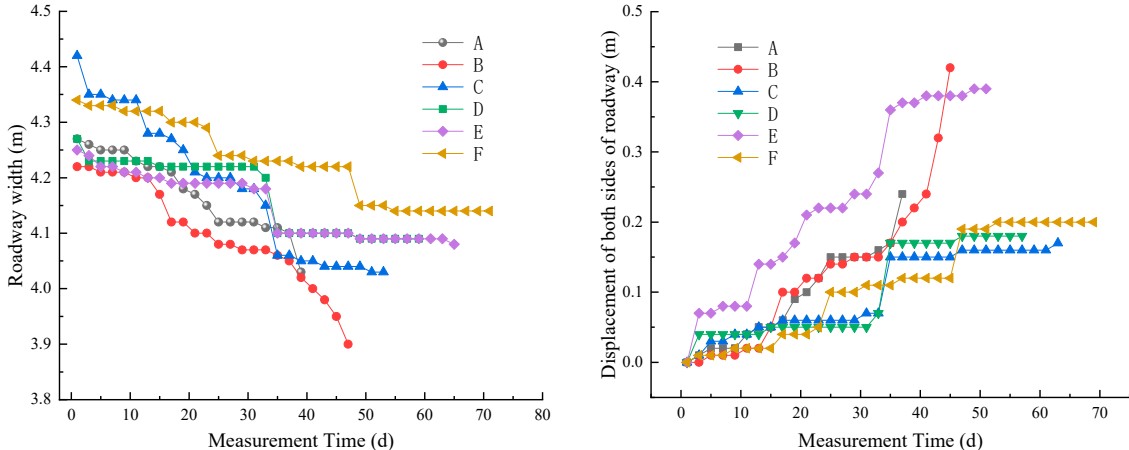

**Figure 12.** Variation curve of distance between two sides of ZF3806 mining roadway.

Figure 13 shows the curve of distance change and displacement change of the roof and floor of the mining roadway on the ZF3806 working face. The changing trend of the top and bottom plate distance at the ABC point of the non-grouting section is the same with large deformation and damage. The closer the roadway is to the working face, the faster the roof will cause subsidence, and with the advancement of the working face, the subsidence speed will accelerate significantly. Three DEF points in the grouting section have a small deformation at the initial stage, which is gradually stable later. The deformation is small, and the overall deformation only occurs near the working face. The deformation difference between the non-grouting section and the grouting section is noticeable. The maximum deformation of the non-grouting section can reach more than 0.53 m, and the maximum deformation of the grouting section is about 0.12 m. After grouting, the roof deformation of the grouted section is small, there is no bolt falling off or roof caving, and the roadway integrity is good. The analysis shows the roof crack in the non-grouting section is developed and has a high degree of fragmentation, which is easily affected by mining. There is obvious deformation and damage during periodic weighting. After grouting, the roof of the grouting section formed an integral structure with good integrity. At the initial stage, because the grouting fluid has not reached its stable strength, the roof has a small amount of displacement, and at the later stage, there is a small amount of overall subsidence under the influence of mining.

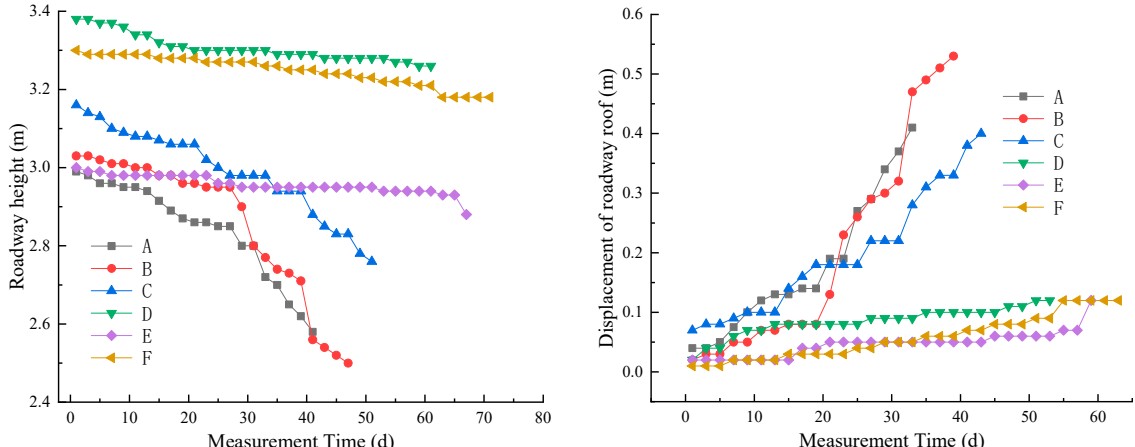

**Figure 13.** Distance variation curve of ZF3806 roof and floor of mining roadway.

As shown in Figure 14, the roadway is stable after the layered grouting method with the core of "shallow bolt grouting + deep cable grouting" is adopted. The roadway surface displacement is reduced, and the maximum roof displacement is 0.12 m. The roof's overall integrity is good, and no roof leakage or bolt is falling off; bolt failure no longer occurs. The later maintenance cost of the roadway is significantly reduced, ensuring the safe and efficient mining of the working face. After layered grouting, the surrounding rock of the roadway can release the bearing stress and maintain the stability of the surrounding rock.

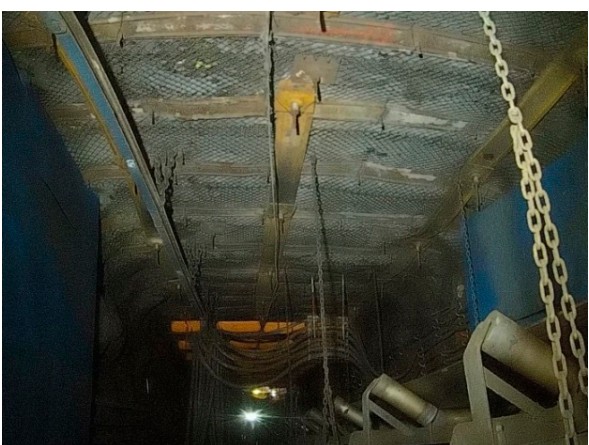

**Figure 14.** Effect of grouting support.

## 6. Conclusions

1. Field investigation and drilling exploration were carried out for the mining roadway of ZF3806's working face, and rock mechanics experiments were carried out. The analysis results show the coal and rock masses have significant fragmentation, the roof subsidence is severe, and the support failures are more frequent. During mining, accidents, such as the roof falling and anchor bolt failure, occur, and there are significant potential safety hazards.

2. The layered grouting reinforcement method of "shallow bolt grouting + deep cable grouting" is proposed, and the bending moment equation of roof rock beam is established. Through layered grouting, the strength of the surrounding rock is increased, and the bending moment of roof rock is reduced so the roadway roof forms a composite beam support structure, and its ability to resist bending deformation is strengthened.

3.　The layered grouting of "shallow bolt grouting + deep cable grouting" is adopted to reinforce the support, and the field application effect is significant. According to the field-monitoring results, the stability of the roadway is improved, the roof displacement is reduced, and the integrity is better. The displacement of both sides of the roadway is within the controllable range, which ensures safe mining and reduces the later maintenance cost of the roadway.

4.　The research of this paper is mainly field oriented. This paper mainly studies the rock mechanics failure characteristics of the roadway roof and the mechanism of grouting reinforcement support and carries out field application. Due to the influence of objective factors, such as experimental conditions, further research will be carried out on the performance of grouting materials and the comparison before and after grouting in broken rock mass.

**Author Contributions:** Writing—original draft preparation, Z.L.; data curation, Z.L. and T.F.; investigation, Z.L. and T.F.; formal analysis, Z.L.; funding acquisition, Z.L. and T.F.; methodology, Z.L. All authors have read and agreed to the published version of the manuscript.

**Funding:** This research was funded by the National Natural Science Foundation of China (51974117, 52174110) and the Natural Science Foundation of Hunan Province (2020JJ4027).

**Institutional Review Board Statement:** Not applicable.

**Informed Consent Statement:** Not applicable.

**Data Availability Statement:** The data used to support the findings of this study are available from the corresponding authors upon request.

**Acknowledgments:** The authors are thankful for the support and help of Zhao Ji from Shanxi Binxian Shuiliandong Coal Mine Co., Ltd.

**Conflicts of Interest:** The authors declare no conflict of interest.

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
