# Peer review of "Mechanism and Application of Layered Grouting Reinforcement for Fractured Coal and Rock Roadway"

_applsci, doi:10.3390/app13020724_

Round 1

Author Response

Thank you for your comments concerning our manuscript entitled " Mechanism and application of layered grouting reinforcement for fractured coal and rock roadway." Those comments are valuable and helpful for revising and improving our paper and the important guiding significance to our research in the future. We have studied comments carefully and have made revisions, which we hope to meet with approval. Revised portions are marked in red in the manuscript. The primary revisions in the manuscript and the response to the comments are as following:

Point 1: The abstract sounds clear but keywords need to be modified. Please use words not combinations of words or phrases. “Fractured coal and rock mass” seems to be too long.

Response 1: Thank you for underlining this deficiency. We have revised the keywords. Please refer to the revised manuscript.

Point 2: You should give more keywords.

Response 2: We appreciate it very much for this good suggestion, and we have done it according to your ideas. Please refer to the revised manuscript.

Point 3: In the Introduction section, an enhanced literature review is required. For this study, the authors have used only predominantly literature sources from China. It seems to be insufficient. Please consider the suggested research (comes from Ukraine, Kazakhstan and VietNam) in your paper when enhancing the introduction section. I believe they are worth considering in your paper.

Krykovskyi, O., Krykovska, V., & Skipochka, S. (2021). Interaction of rock-bolt supports while weak rock reinforcing by means of injection rock bolts. Mining of Mineral Deposits, 15(4), 8-14. doi:10.33271/mining15.04.008

Begalinov, A., Almenov, T., Zhanakova, R., & Bektur, B. (2020). Analysis of the stress deformed state of rocks around the haulage roadway of the Beskempir field (Kazakhstan). Mining of Mineral Deposits, 14(3), 28-36. doi:10.33271/mining14.03.028

Vu, T.T (2022). Solutions to prevent face spall and roof falling in fully mechanized longwall at underground mines, Vietnam. Mining of Mineral Deposits, 16(1), 127-134. doi:10.33271/mining16.01.127

Response 3: We appreciate it very much for this good suggestion, and we have done it according to your ideas. Please refer to the revised manuscript.

Point 4: You have indicated in the Introduction section – The study's findings could be useful in treating broken surrounding rock in other coal mine roadways. So, the question is how the expected result be used or implemented within other geological conditions? What limitations?

Response 4: Thank you for your comment. Grouting reinforcement has been successfully applied in many coal mine roadway supports, and the introduction of this paper has also mentioned. We believe that layered grouting can play a good role in strengthening the broken rock mass, especially for coal mine roadways with obvious roof layering and more bedding.

Point 5: Something wrong with the Columnar lithology. Why it is the same for the different lithology?

Response 5: Thank you for your comment. Different lithology are different. Siltstone is four dots, Fine sandstone is three dots, and silty fine sandstone is three dots alternating with four dots.

Point 6: The title of the Figure 5 is incorrect (Plastic deformation zone of roadway surrounding rock). Here is shown not only the plastic deformation zone. It must be renamed.

Response 6: Thank you for your valuable and thoughtful comments. We have modified the title of Figure 5 as the distribution of roadway surrounding rock state. Please refer to the revised manuscript.

Point 7: Elastic, plastic and fracture regions or zones?

Response 7: Thank you for your valuable and thoughtful comments. After consulting literature and dictionaries, we use zones uniformly. Please refer to the revised manuscript.

Point 8: Figure 6 Grouting diagram. Can you provide some sizes?

Response 8: Thank you for your comment. Figure 6 is a schematic diagram of grouting. Different working conditions will lead to different situations, so a specific size cannot be provided.

Point 9: Please provide a short description of further research.

Response 9: We appreciate it very much for this good suggestion. We have added a part of discussion on the further research contents in the conclusion. The contents added are as follows: “The research of this paper is mainly field oriented. This paper mainly studies the rock mechanics failure characteristics of the roadway roof and the mechanism of grouting reinforcement support, and carries out field application. Due to the influence of objective factors such as experimental conditions, further research will be carried out on the performance of grouting materials and the comparison before and after grouting in broken rock mass.” Please refer to the revised manuscript.

Point 10: The novelty of the paper must be highlighted in the conclusions section.

Response 10: Thank you for your valuable and thoughtful comments. We have revised the Conclusion section. Please refer to the revised manuscript.

Point 11: In general, I must admit that a very good study was performed, and I will recommend your paper for publication after careful revision. The content of the manuscript is like that of a case study. The knowledge contained here may be useful for engineers, students, and scientists, searching for any knowledge related to mining engineering, which is the most important value of the manuscript.

Response 11: Thank you very much for your valuable comments on this paper. We have revised all the comments and marked them in red.

Once again, we thank the Editors & Reviewers for the time you put in reviewing our manuscript. Your comments have been precious.

Reviewer 2 Report

In this study, the authors investigated the mechanism and strategies of the reinforcement for coal and rock roadway. The manuscript although has showed some interesting results and is documented with adequate literature references; the novelty of this study should be emphasized.

1.     The novelty of this study should be emphasized. In the abstract, the authors stated that “The study’s findings could be useful in the treating broken surrounding rock in other coal mine roadways”. However, the contributions should be emphasized clearly and included in the Conclusions.

2.     As for section 3.2, the information of standard specimens should be included.

3.     Fig.5 demonstrated that different zones. It seems that the particle size in Facture region is much larger than the other regions. Does it correspond to the site conditions?

4.     Fig.5 and Fig. 8, there are different zones in the roof. There are two supports at each side and no support in the middle of the beam. Why is the thickness of each zone same no matter at side and in the middle? 

5.  Writing should be improved. Please correct the errors in the manuscripts and keep the consistency of tenses.

Author Response

Thank you for your letter and the reviewers' comments concerning our manuscript entitled " Mechanism and application of layered grouting reinforcement for fractured coal and rock roadway." Those comments are valuable and helpful for revising and improving our paper and the important guiding significance to our research in the future. We have studied comments carefully and have made revisions, which we hope to meet with approval. Revised portions are marked in red in the manuscript. The primary revisions in the manuscript and the response to the comments are as following:

Point 1: The novelty of this study should be emphasized. In the abstract, the authors stated that “The study’s findings could be useful in the treating broken surrounding rock in other coal mine roadways”. However, the contributions should be emphasized clearly and included in the Conclusions.

Response 1: Thank you for your valuable and thoughtful comments. We have revised the conclusion section and emphasized the novelty and contributions of this paper. Please refer to the revised manuscript.

Point 2: As for section 3.2, the information of standard specimens should be included.

Response 2: Thank you for underlining this deficiency. We have added the following: According to the research content, the test rock sample is taken from ZF3806 working face of Shuiliandong Coal Mine in Bin County, Shanxi Province. The rock samples are roof coal, mudstone, silty fine sandstone and siltstone. Through on-site sampling, the rock samples are transported from the underground to the ground, and coal cores and rock cores are drilled in the laboratory with coring machine; Then cut and polish the end with an end grinder. All standard specimens prepared are Φ 50mm × 100mm, standard specimens processing equipment and process are shown in the figure 4. Please refer to the revised manuscript.

Point 3: Fig.5 demonstrated that different zones. It seems that the particle size in Facture region is much larger than the other regions. Does it correspond to the site conditions?

Response 3: Thank you for your comment. Figure 5 is a simplified diagram, which is used to show the different zoning states of the surrounding rock of the roadway.

Point 4: Fig.5 and Fig. 8, there are different zones in the roof. There are two supports at each side and no support in the middle of the beam. Why is the thickness of each zone same no matter at side and in the middle?

Response 4: Thank you for your comment. In Figure 8, the supports on both sides are the two sides of the roadway. There is no support in the middle part, and the bolts and cables support are simplified as support resistance P1 and P2. At the same time, the thickness of simply supported beam does not represent the thickness in the actual project, but is a simplification.

Point 5: Writing should be improved. Please correct the errors in the manuscripts and keep the consistency of tenses.

Response 5: We appreciate it very much for this good suggestion, and we have done it according to your ideas. Please refer to the revised manuscript.

Once again, we thank the Editors & Reviewers for the time you put in reviewing our manuscript. Your comments have been precious.

Round 2

Reviewer 1 Report

Dear authors,

I am satisfied with the corrections provided by you.

This study is an important contribution to sustainable mining.

Congratulations to the authors.